# Crosstalk between the Circadian Clock and Histone Methylation

**DOI:** 10.3390/ijms23126465

**Published:** 2022-06-09

**Authors:** Changhui Sun, Shihang Liu, Changcai He, Chao Zhong, Hongying Liu, Xu Luo, Ke Li, Kuan Zhang, Qian Wang, Congping Chen, Yulin Tang, Bin Yang, Xiaoqiong Chen, Peizhou Xu, Ting Zou, Shuangcheng Li, Peng Qin, Pingrong Wang, Chengcai Chu, Xiaojian Deng

**Affiliations:** 1State Key Laboratory of Crop Gene Exploration and Utilization in Southwest China, Rice Research Institute, Sichuan Agricultural University, Chengdu 611130, China; lsh2372728743@163.com (S.L.); he15680693552@163.com (C.H.); 18402877732@163.com (C.Z.); l18892802593@163.com (H.L.); lx2434340736@163.com (X.L.); keli2251630252@163.com (K.L.); zhangkuan7@hotmail.com (K.Z.); wq1315522095@gmail.com (Q.W.); cpchen2501@163.com (C.C.); tyl18892891705@163.com (Y.T.); yb18080897532@163.com (B.Y.); xiaochenq777@126.com (X.C.); xpzhxj@163.com (P.X.); zouting@sicau.edu.cn (T.Z.); lisc926105@163.com (S.L.); qinpeng626@gmail.com (P.Q.); prwang@sicau.edu.cn (P.W.); 2State Key Laboratory for Conservation and Utilization of Subtropical Agro-Bioresources, College of Agriculture, South China Agricultural University, Guangzhou 510642, China; ccchu@genetics.ac.cn

**Keywords:** rice (*Oryza sativa*), *OsLHY*, *SDG724*, circadian clock, histone modification, crosstalk, transcriptome analysis

## Abstract

The circadian clock and histone modifications could form a feedback loop in Arabidopsis; whether a similar regulatory mechanism exists in rice is still unknown. Previously, we reported that *SDG724* and *OsLHY* are two rice heading date regulators in rice. *SDG724* encodes a histone H3K36 methyltransferase, and *OsLHY* is a vital circadian rhythm transcription factor. Both could be involved in transcription regulatory mechanisms and could affect gene expression in various pathways. To explore the crosstalk between the circadian clock and histone methylation in rice, we studied the relationship between *OsLHY* and *SDG724* via the transcriptome analysis of their single and double mutants, *oslhy*, *sdg724*, and *oslhy*
*sdg724*. Screening of overlapped DEGs and KEGG pathways between *OsLHY* and *SDG724* revealed that they could control many overlapped pathways indirectly. Furthermore, we identified three candidate targets (*OsGI*, *OsCCT38,* and *OsPRR95*) of OsLHY and one candidate target (*OsCRY1a*) of SDG724 in the clock pathway. Our results showed a regulatory relationship between *OsLHY* and *SDG724*, which paved the way for revealing the interaction between the circadian clock and histone H3K36 methylation.

## 1. Introduction

The circadian clock is an endogenous rhythmic oscillation that controls many plant life processes, such as flowering time, germination, growth, floral scent emission, flower opening, and so on [1,2]. In plants, the circadian clock is formed by multiple interlocked transcriptional feedback loops. In Arabidopsis, the feedback loops are constructed by more than 20 clock factors [1,3]. Among them, *CIRCADIAN CLOCK ASSOCIATED 1* (*CCA1*), *LATE ELONGATED HYPOCOTYL* (*LHY*), and *TIMING OF CAB EXPRESSION 1* (*TOC1*) form the core loop and are responsible for the robust rhythm activity [4].

In eukaryotic cells, genomic DNA is packaged with histones and forms the repeating nucleosome unit of the chromatin. The modifications in the histone N-terminal tail, which control chromatin structure, are directly linked to transcriptional regulation and are one of the most important parts of epigenetics [3,5]. In Arabidopsis, recent studies show that the circadian clock genes can directly regulate histone modification regulators. These modifications, in turn, react to the circadian clock, forming a mutual regulation in circadian clock mechanism [2,3,6]. However, such a relationship underlying the epigenetic modification and circadian clock is poorly understood in rice.

Previously, we reported a histone H3 lysine 36 (H3K36) methyltransferase SET Domain Group protein 724 (SDG724). The mutant *sdg724* (also called *lvp1*) exhibited a later flowering phenotype under long-day (LD) and short-day (SD) conditions. Our study proposed an epigenetic regulation mechanism for rice flowering that SDG724 mediates *Oryza sativa MADS50* (*OsMADS50*) and *RICE FLOWERING LOCUS T1* (*RFT1*) loci H3K36me2/3 deposition to active flowering transition [7]. Subsequently, by mutagenesis of *sdg724*, we identified a later flowering *lvp1 enhancer mutant 1* (*lem1*). *LEM1* was identified as the core clock factor *OsLHY* [4]. Interestingly, *OsLHY* promotes flowering under ≥12-h day length but inhibits heading under ≤11-h day length. Our study further exhibited that OsLHY could inhibit *Oryza sativa*
*GIGANTEA* (*OsGI*) expression by binding to the CCA1-binding sequence (CBS) element on its promoter and regulating heading date through upregulating *Hd1* expression. We also designed a series of transcriptome analyses for *OsLHY* function investigation, and seedlings of Nipponbare (wild type, WT), *oslhy*, *sdg724,* and the double mutant *oslhy sdg724* were collected for RNA-sequencing. KEGG (Kyoto Encyclopedia of Genes and Genomes, https://www.genome.jp/kegg/, accessed around 6 April 2022) pathway enrichment showed that the circadian rhythm was the most significantly affected pathway of *OsLHY*, implying that *OsLHY* was a critical clock component. In this study, these data were further analyzed to explore the interaction between the circadian clock and histone methylation. To obtain the most real regulatory relationship, we used a very strict threshold (padj < 0.01) for differentially expressed gene (DEG) detection [4]. Our results exhibited that although OsLHY and SDG724 might not tend to regulate the same targets directly, they could control numbers of the same pathways indirectly, implying an interesting regulatory relationship between the core clock pathway and histone H3K36 methylation.

## 2. Results

### 2.1. SDG724 Has a Huge Influence on the Function of OsLHY

To determine the relationship between *SDG724* and *OsLHY*, we firstly checked the transcription varies in WT vs. *oslhy* and *sdg724* vs. *oslhy sdg724* comparisons. There were 528 DEGs in WT vs. *oslhy* comparison and 272 DEGs in the *sdg724* vs. *oslhy sdg724* comparison (Figure 1A). As the DEG quantity was significantly reduced in the *oslhy sdg724* background, it seems that *sdg724* has a huge influence on the function of *OsLHY*. We also conducted a KEGG enrichment analysis of these DEGs. The results revealed that the DEGs of WT vs. *oslhy* comparison were enriched in 83 categories, and the DEGs of *sdg724* vs. *oslhy sdg724* comparison were only enriched in 47 pathways. However, 41 pathways out of the *sdg724* vs. *oslhy sdg724* comparison were merged with the pathways of WT vs. *oslhy* comparison, and only six pathways uniquely existed in the *sdg724* vs. *oslhy sdg724* comparison (Figure 1B).

Furthermore, we identified 36 over-lapping DEGs between the two comparisons enriched in nine KEGG pathways: circadian rhythm, thiamine metabolism, phenylpropanoid biosynthesis, biosynthesis of cofactors, oxidative phosphorylation, MAPK signaling pathway, starch and sucrose metabolism, plant-pathogen interaction, and plant hormone signal transduction. Certainly, the circadian rhythm was the most significant pathway, implying again that *OsLHY* was a vital circadian rhythm factor (Figure 1C) [4].

### 2.2. OsLHY Has a Huge Influence on the Function of SDG724

As a global H3K36 methyltransferase, SDG724 might contribute to multiple regulatory pathways in rice [7]. To further identify more functions of SDG724, we conducted two comparisons [4]. In the WT vs. *sdg724* comparison, there were 750 DEGs; in the *oslhy* vs. *oslhy sdg724* comparison, there were 220 DEGs. We identified 105 overlapped DEGs in the two comparisons (Figure 2A). As the DEGs sharply decreased in the *oslhy* vs. *oslhy sdg724* comparison, it seems that the *oslhy* background has a huge influence on the function of *SDG724*.

KEGG enrichment analysis revealed that the DEGs of the WT vs. *sdg724* comparison were enriched in 82 pathways, and the DEGs of *oslhy* vs. *oslhy sdg724* comparison were enriched in 40 pathways. Interestingly, 37 pathways overlapped between the two comparisons, and only three uniquely existed in the WT vs. *sdg724* comparison (Figure 2B).

Furthermore, the overlapped 105 DEGs between the two comparisons were enriched in 20 KEGG pathways. The top ten were the phosphatidylinositol signaling system, homologous recombination, fatty acid metabolism, taurine and hypotaurine metabolism, butanoate metabolism, mismatch repair, beta-alanine metabolism, fatty acid biosynthesis, DNA replication, and alanine, and aspartate and glutamate metabolism (Figure 2A). However, the circadian rhythm was not found in all 20 KEGG pathways, implying again that *SDG724* was not a major circadian rhythm factor [4].

### 2.3. Exploring the Targets of OsLHY in the Clock Pathway

As a repressive transcriptional regulator, some upregulated DEGs in *oslhy* could be the direct targets of OsLHY. We have reported that the circadian rhythm was the most significantly affected pathway in the upregulated DEGs of the WT vs. *oslhy* and *sdg724* vs. *oslhy sdg724* comparisons [4]. To further identify more reliable candidate target genes of OsLHY, we conducted an overlap analysis between the upregulated DEGs in the two comparisons. As a result, 22 genes were identified and were enriched in seven KEGG pathways (Figure 3A,B). The circadian rhythm was also the most significantly affected pathway, including three genes, *OsGI*, *Pseudo*-*Response Regulator 95* (*OsPRR95*), and *OsCCT38*, which were further validated by quantitative real-time RT-PCR (qRT-PCR) (Figure 3C).

It has been reported that decreased *OsGI* expression by RNAi could lead to early flowering under LD conditions and late flowering under SD conditions in rice [8]. In our previous study, *OsGI* was proved to be the target of OsLHY, implying our RNA-sequencing analysis was reliable [4]. Recently, *OsCCT38* was confirmed as a bifunctional flowering gene in rice [9]. Knockout of the *OsCCT38* mutant showed earlier heading under LD conditions and later heading under SD conditions. If OsLHY could directly regulate *OsCCT38*, *OsCCT38* expression would be upregulated in *oslhy* and thus induce later flowering under LD and earlier flowering under SD conditions, which is consistent with the *oslhy* flowering phenotype. *OsPRR95*, a homologous gene of Arabidopsis clock regulator *PRR5*, has not been cloned and could serve as another candidate target gene of OsLHY for further study [10].

### 2.4. Exploring the Targets of SDG724 in the Clock Pathway

SDG724 functions as a major H3K36 methyltransferase and activates gene expression in rice. To further identify the possible direct targets of SDG724, the downregulated DEGs of the WT vs. *sdg724* comparison were analyzed. As a result, there were 368 downregulated DEGs enriched in 62 KEGG pathways (Figure 4A,B). Moreover, only one DEG was found in the circadian rhythm pathway and was identified as a rice blue-light photoreceptor *CRYPTOCHROME 1* (*OsCRY1a*) (Appendix A). *OsCRY1a* is involved in leaf and coleoptile elongation in the early seedling stage [11,12]. According to the Erice epigenomic platform (http://www.elabcaas.cn/rice/index.html, accessed around 16th April, 2022), the chromatin loci around *OsCRY1a* has many histone modifications, including H3K36me1/2/3 (Figure 4B), suggesting that SDG724 might regulate these H3K36 methylations and improve *OsCRY1a* expression. The qRT-PCR results validated the higher expression in *sdg724* than in WT (Figure 4C). In the future, we need to check the blue light-sensitive ability of *sdg724* and figure out the relationship between *SDG724* and *OsCRY1a*.

In the *oslhy* vs. *oslhy sdg724* comparison, there were 119 downregulated DEGs enriched in 21 KEGG pathways (Figure 4A, Appendix A). Compared with 62 KEGG pathways in the WT vs. *sdg724* comparison, only one unique pathway was found, and 20 pathways were merged (Figure 4B). Furthermore, we did not find any DEGs enriched in the circadian rhythm pathway, suggesting that *OsCYR1a* was not affected in the *oslhy* mutant background.

Furthermore, there were 70 overlapped downregulated DEGs between the two comparisons, enriched in 12 pathways, such as the phosphatidylinositol signaling system, homologous recombination, fatty acid metabolism, mismatch repair, fatty acid biosynthesis, DNA replication, alpha-Linolenic acid metabolism, fatty acid degradation, inositol phosphate metabolism, nucleotide excision repair, glutathione metabolism, and spliceosome, implying a multifunction of *SDG724*.

### 2.5. OsLHY and SDG724 Might Not Share the Same Target

In Arabidopsis, CCA1/LHY can recruit a histone modification complex to their target loci for transcription regulation through histone modifications [6]. OsLHY is a repressive transcription factor [4], while SDG724 acts as a major H3K36 methyltransferase for transcriptional activation [7]. To identify the potential targets of both OsLHY and SDG724, we conducted an overlap analysis between 313 upregulated DEGs of WT vs. *oslhy* and 368 downregulated DEGs of WT vs. *sdg724* (Figure 5A). Interestingly, we could not find any overlapped genes, implying that OsLHY and SDG724 might not work together on the same downstream target (Figure 5A). Furthermore, our yeast two-hybrid assay verified that OsLHY and SDG724 could not interact with each other (Figure 5B). Therefore, based on our current data, OsLHY and SDG724 might not share the same target.

### 2.6. Most of the Pathways Controlled by OsLHY and SDG724 Merged

Although OsLHY and SDG724 might not regulate the same downstream target genes, 121 DEGs between WT vs. *oslhy* and WT vs. *sdg724* overlapped (Figure 6A), implying that *OsLHY* and *SDG724* could be involved in these DEGs indirectly. KEGG enrichment analysis revealed that 65 pathways overlapped between the two comparisons (Figure 6B). Interestingly, only 18 unique pathways belonged to the WT vs. *oslhy* comparison, and only 17 pathways uniquely belonged to the WT vs. *sdg724* comparison, suggesting that *OsLHY* and *SDG724* share most of the downstream KEGG pathways (Figure 6B).

Furthermore, the overlapped 121 DEGs were enriched in 30 KEGG pathways (Figure 6C). The top ten were alanine, aspartate and glutamate metabolism; fructose and mannose metabolism; biosynthesis of amino acids; one carbon pool by folate; amino sugar and nucleotide sugar metabolism; mismatch repair; phenylalanine metabolism; DNA replication; cyanoamino acid metabolism; and alpha-Linolenic acid metabolism. To sum up, *OsLHY* and *SDG724* can cooperate to control many biological processes indirectly in rice.

## 3. Discussion

In our previous study, double mutant *oslhy osgi* analysis exhibited that *OsGI* was not the only target of OsLHY, as *osgi* could not truncate all the signals of *OsLHY*. There must be other targets of OsLHY [4]. *OsLHY* performs a dual role in rice flowering transition, so flowering regulators possessing such dual functions are likely in a relationship with *OsLHY*. Recently, *OsCCT38* was identified as a bifunctional flowering factor, suppressing heading under LD conditions and promoting heading under SD conditions [9]. *OsCCT38* was a significant DEG in both the WT vs. *oslhy* and *sdg724* vs. *oslhy sdg724* comparisons in our study. Therefore, *OsCCT38* might be another downstream target gene in setting a certain daylength threshold for triggering photoperiodic flowering.

As a core clock regulator, *OsLHY* can control many biological processes. Recently, it has been reported to be involved in various pathways, including tiller growth, panicle development, nitrogen-mediated flowering time, ABA signaling, and so on [13,14,15,16]. Our transcriptome analyses in this study will help determine more biological functions of *OsLHY*.

After our last paper [7], there were few reports of *SDG724*. However, *sdg724* also showed multiply phenotypes in rice, such as higher plant height, longer panicle length, lower germination rate, etc. Exploring the target genes of SDG724 is beneficial for elucidating the regulatory mechanisms of these phenotypes. *OsCRY1a* has been identified as a blue-light receptor for rice leaf and coleoptile elongation [12]. Considering that the expression of *OsCRY1a* was downregulated in *sdg724* and that its chromosome loci possess histone-modified H3K36me1/2/3 peaks, SDG724 might deposit the methylated H3K36 on *OsCRY1a* chromatin and improve its expression. However, more studies are needed to determine the relationship between *SDG724* and *OsCRY1*.

Recent studies showed that the circadian clock and histone modifications could form a feedback loop for transcription regulation in Arabidopsis [2,3]. However, such a regulatory mechanism is still unknown in rice. In our study, there were no overlapped genes between the upregulated DEGs of WT vs. *oslhy* and the downregulated DEGs of WT vs. *sdg724*, indicating that SDG724 and OsLHY might not work together on the same target genes. However, *OsLHY* and *SDG724* had a huge influence on the function of each other, and most of the KEGG pathways controlled by *SDG724* and *OsLHY* merged. We demonstrated some relationships between *OsLHY* and *SDG724*, laying a good foundation for revealing the crosstalk between the circadian clock and histone methylation (Figure 7).

Additionally, it needs to be pointed out that *OsLHY* and *SDG724* are constitutive expressions, and their proteins are assumed to be localized in the nucleus, providing a crosstalk possibility in space. However, the samples we collected for RNA sequencing were whole seedlings with a mixture of different kinds of cells. As interactions between genes or proteins can occur in specific cells, transcriptome analysis without distinguishing cell fate and epigenetic status could not 100% reflect the reality. Further, DEGs might also be specifically enriched in KEGG pathways in different cell types. Thus, further transcriptome analysis in different tissues or single cells and detailed chromatin analysis are on our schedule, hoping to find more interesting scientific phenomena between *OsLHY* and *SDG724*.

## 4. Materials and Methods

### 4.1. RNA Sequencing and Analysis

In this study, all of the RNA sequencing data were from our last paper [4]. WT (Nipponbare), *oslhy*, *sdg724*, and the double mutant *oslhy sdg724* were planted in an artificial chamber under 12-h light/12-h dark conditions. Two-week-old seedlings were collected 2 h after dawn for RNA sequencing [4].

### 4.2. DEGs and KEGG Pathway Investigation

KEGG pathway analysis of DEGs was carried out using the Kyoto Encyclopedia of Genes and Genomes (KEGG) database [17], and KEGG enrichment was done according to the Clusterprofiler method [18]. The results of the KEGG enrichment analysis used ggplot2 (version 3.3.5) to display results in a bubble chart [19]. All R packages were installed via Bioconductor (version 1.30.16) [20]. Overlapping diagrams were produced by VennDiagrams pot (http://www.ehbio.com/ImageGP, accessed around 13 April 2022) [21].

### 4.3. qRT-PCR Analysis

The seedling samples were the same as those used in RNA-sequencing. Total RNA was extracted by TRIzol reagent (R401-01, Vazyme, Nanjing, China). Using a reverse transcription kit (R133-01, Vazyme), first-strand cDNA was synthesized from an RNA template. qRT-PCR was performed using ChamQ Universal SYBR qPCR Master Mix (Q711-03, Vazyme) by a CFX96 real-time PCR system (Bio-Rad, Hercules, CA, USA) [4]. All primers used for qRT-PCR are listed in Appendix A.

### 4.4. Y2H Assay

The full-length coding sequences of OsLHY and SDG724 were cloned into pGADT7 and pGBKT7, respectively. These two vectors and positive vectors were cotransformed into the yeast strain Y2H gold [22]. All primers used here are listed in Appendix A.

## Figures and Tables

**Figure 1 ijms-23-06465-f001:**
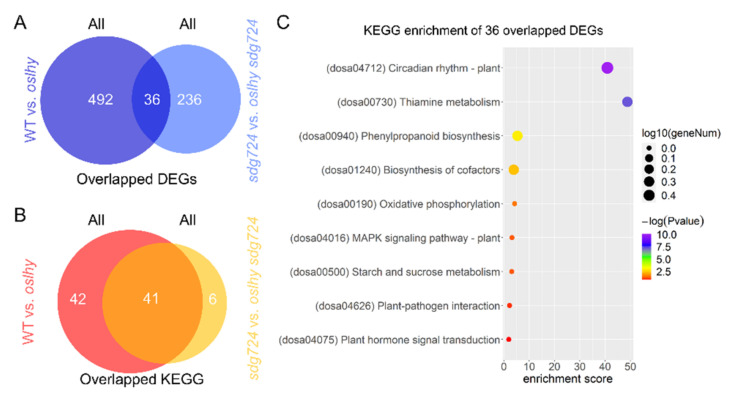
The influence of *SDG724* on *OsLHY*. (**A**) Overlapped DEGs between the WT vs. *oslhy* and *sdg724* vs. *oslhy sdg724* comparisons. (**B**) Overlapped KEGG pathways between the WT vs. *oslhy* and *sdg724* vs. *oslhy sdg724* comparisons. (**C**) KEGG enrichment of 36 overlapped DEGs between the WT vs. *oslhy* and *sdg724* vs. *oslhy sdg724* comparisons. All indicates all of the DEGs.

**Figure 2 ijms-23-06465-f002:**
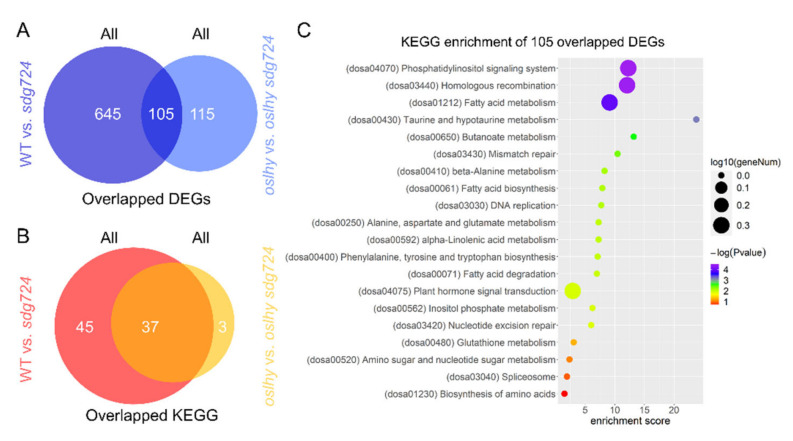
The influence of *OsLHY* on *SDG724*. (**A**) Overlapped DEGs between the WT vs. *sdg724* and *oslhy* vs. *oslhy sdg724* comparisons. (**B**) Overlapped KEGG pathways between the WT vs. *sdg724* and *oslhy* vs. *oslhy sdg724* comparisons. (**C**) KEGG enrichment of 105 overlapped DEGs between the WT vs. *sdg724* and *oslhy* vs. *oslhy sdg724* comparisons. All indicates all of the DEGs.

**Figure 3 ijms-23-06465-f003:**
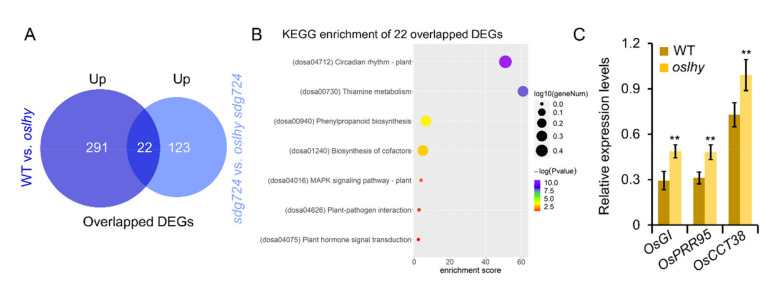
Candidate targets analysis of OsLHY. (**A**) Overlapped upregulated DEGs between WT vs. *oslhy* and *sdg724* vs. *oslhy sdg724* comparisons. Up indicates the upregulated DEGs. (**B**) KEGG enrichment of 22 overlapped upregulated DEGs between WT vs. *oslhy* and *sdg724* vs. *oslhy sdg724* comparisons. (**C**) qRT-PCR validation of the candidate target genes in the clock pathway. Asterisks indicate statistically significant differences by a Student’s *t*-test (** *p* < 0.01).

**Figure 4 ijms-23-06465-f004:**
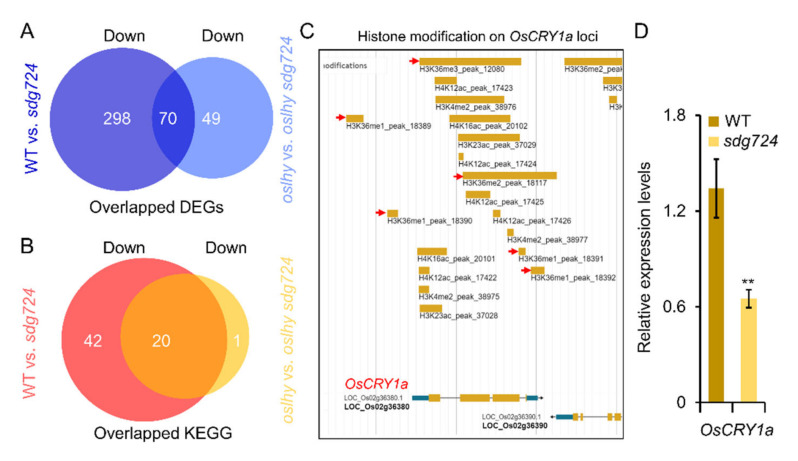
Candidate targets analysis of SDG724. (**A**) Overlapped downregulated DEGs between WT vs. *sdg724* and *oslhy* vs. *oslhy sdg724* comparisons. (**B**) Overlapped KEGG pathways between WT vs. *sdg724* and *oslhy* vs. *oslhy sdg724* comparisons. (**C**) Histone modifications around *OsCRY1a* loci. The data is from Erice epigenomic platform (http://www.elabcaas.cn/rice/index.html, accessed around 16 April 2022). (**D**) qRT-PCR validation of candidate target genes in clock pathway. Down indicates the downregulated DEGs. Asterisks indicate statistically significant differences by a Student’s *t*-test (** *p* < 0.01).

**Figure 5 ijms-23-06465-f005:**
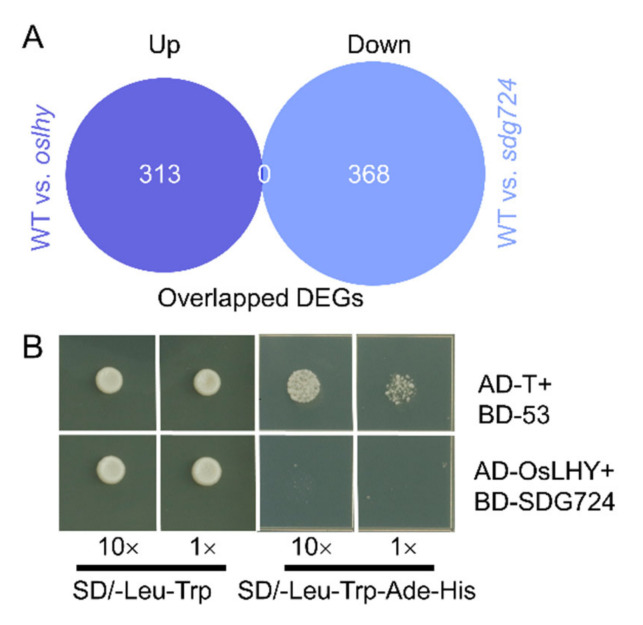
No physical interaction between OsLHY and SDG724 proteins. (**A**) Overlap analysis between upregulated DEGs of WT vs. *oslhy* and downregulated DEGs of WT vs. *sdg724*. Up indicates the upregulated DEGs; down indicates the downregulated DEGs. (**B**) Yeast two-hybrid assays (Y_2_H) showed that OsLHY could not interact with SDG724. Co-transformation of AD-T and BD-5 was used as the positive control.

**Figure 6 ijms-23-06465-f006:**
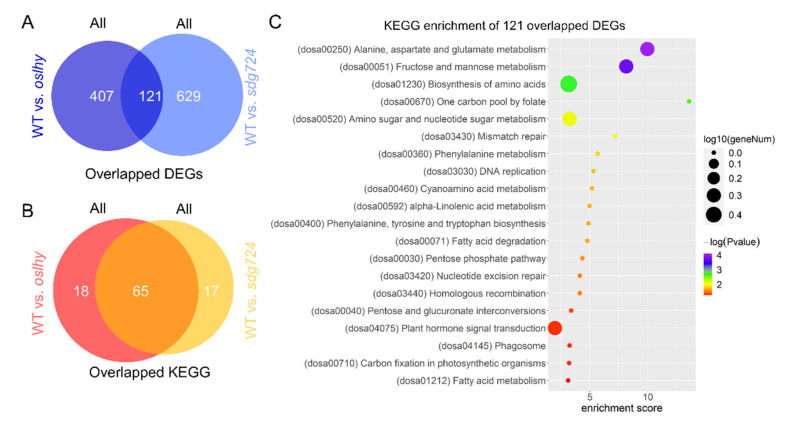
Overlapped pathways controlled by *OsLHY* and *SDG724*. (**A**) Overlapped DEGs between the WT vs. *oslhy* and WT vs. *sdg724* comparisons. (**B**) Overlapped KEGG pathways between the WT vs. *oslhy* and WT vs. *sdg724* comparisons. (**C**) KEGG enrichment of 121 overlapped DEGs between the WT vs. *oslhy* and WT vs. *sdg724* comparisons.

**Figure 7 ijms-23-06465-f007:**
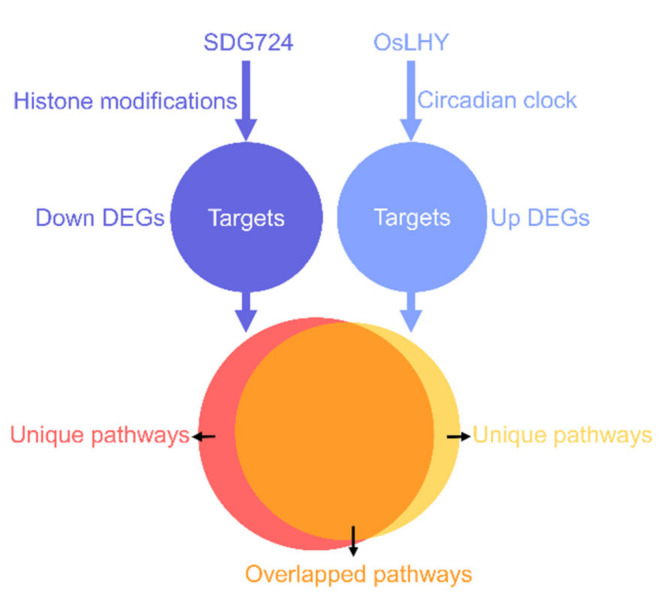
A simple model of the relationship between *OsLHY* and *SDG724*. SDG724 and OsLHY might not work together on the same target gene. However, most of the KEGG pathways indirectly controlled by *SDG724* and *OsLHY* merged, paving the way for revealing the interaction between the circadian clock and histone modifications.

## Data Availability

Not applicable.

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
