# Peer review of "Crosstalk between the Circadian Clock and Histone Methylation"

_ijms, 2022, doi:10.3390/ijms23126465_

Round 1

Reviewer 1 Report

This is a very interesting paper with nice approach. Authors shown that OsLHY and SDG724 might not tend to regulate the same targets directly, but they control many same pathways indirectly, implying an important regulatory relationship between histone H3K36 methylation and circadian clock.

The paper well-written but required some corrections.

Detailed comments:

The main problem is the experiment setting. Histone methylation (H3K36) is cell-fate (position) dependent, and circadian clock is also regulated differently in different cell files. However, RNA was isolated from whole plants and this important points are missing.

Gene/protein-protein  interactions can occurred in specific cell. That’s why total analysis can not 100% reflected reality.

This fact must be related at least in discussion and must be proven in future.

It will be necessary also to discuss abundance of different pathways in different cell types. Please, try to analysis this information.

Line 19: „histone epigenetic modifications“?? either histone modification or epigenetic.

Line 35: „epigenetic genetics“ ??

Line 37: „In Arabidopsis, recent studies show that the circadian clock can 46 directly regulate histone modifications“ – clock itself can not regulate histone modification.

Line 85: „biosynthesis of cofactors“ ??

Author Response

Point-to-point Response

Reviewer: 1

Comments and Suggestions for Authors

This is a very interesting paper with nice approach. Authors shown that OsLHY and SDG724 might not tend to regulate the same targets directly, but they control many same pathways indirectly, implying an important regulatory relationship between histone H3K36 methylation and circadian clock.

The paper well-written but required some corrections.

Detailed comments:

The main problem is the experiment setting. Histone methylation (H3K36) is cell-fate (position) dependent, and circadian clock is also regulated differently in different cell files. However, RNA was isolated from whole plants and this important points are missing. Gene/protein-protein interactions can occurred in specific cell. That's why total analysis can not 100% reflected reality. This fact must be related at least in discussion and must be proven in future.

It will be necessary also to discuss abundance of different pathways in different cell types. Please, try to analysis this information.

Response: Thank you very much for your great comments. We quite agree with you that circadian clock and histone methylation could function in different types of cells. However, our current data analysis illustrates the crosstalk between OsLHY and SDG724 in 2-week-old seedlings, though such crosstalk might be in cells of different types and states. The scientific issues you care about are the direction of our following research. In the future, we will carry out RNA-sequence in different tissues or single cells for further transcriptome analysis, hoping to find more interesting scientific phenomena between histone methylation and circadian clock.

Based on your helpful suggestion, we discussed the crosstalk between circadian clock and histone methylation in different cell files, and the abundance of different pathways in different cell types in the discussion section. Please see Line 245-252.

Line 19: "histone epigenetic modifications"?? either histone modification or epigenetic.

Response: Done. Thank you for pointing this out, and "histone modifications" is used here.

Line 35: "epigenetic genetics"??

Response: Done. Thanks for pointing this out. It should be "epigenetics".

Line 37: "In Arabidopsis, recent studies show that the circadian clock can 46 directly regulate histone modifications" – clock itself can not regulate histone modification.

Response: Good suggestion. We modified the sentence as follows, "In Arabidopsis, recent studies show that the circadian clock genes can directly regulate histone modification regulators."

Line 85: "biosynthesis of cofactors "??

Response: Thank you for pointing this out. "Biosynthesis of cofactors" is a KEGG Pathway (ko01240). A cofactor is a non-protein chemical compound or metallic ion required for an enzyme's role as a catalyst.

Reviewer 2 Report

In this article, the authors explored into crosstalk between two important regulators, SDG724, that is involved in Histone H3K36 methyltransferases and OsLHY, a circadian rhythm transcription factor. They did so by providing the transcriptome analysis of their single and double mutants of oslhy, sdg724 and oshly sdg724. Upon screening the DEGs and KEGG pathways, they found that they may regulate the genes/pathways involved in the process indirectly.

The manuscript is well written and the results are supported by the transcriptome analysis. They have also summerised the findings in a figure, which always help the reader to have the bottomline of the manuscript.

Author Response

Point-to-point Response

Reviewer: 2

Comments and Suggestions for Authors

In this article, the authors explored into crosstalk between two important regulators, SDG724, that is involved in Histone H3K36 methyltransferases and OsLHY, a circadian rhythm transcription factor. They did so by providing the transcriptome analysis of their single and double mutants of oslhy, sdg724 and oshly sdg724. Upon screening the DEGs and KEGG pathways, they found that they may regulate the genes/pathways involved in the process indirectly.

The manuscript is well written and the results are supported by the transcriptome analysis. They have also summerised the findings in a figure, which always help the reader to have the bottomline of the manuscript.

Response: Thank you very much. We appreciate your positive comments.

Round 2

Reviewer 1 Report

Thank you very much for response.

It is great that you plan to study cell-types specifci gene expression and histone modification. I am not sure methods like RNA-sequence in different tissues or single cells for further transcriptome analysis can help too much. The only way to clarify link between circadian clock and epigenetic is in situ by using, for example, antibody (https://www.thermofisher.com/antibody/product/H3K36me2-Antibody-Polyclonal/600-401-I87) and detailed chromatin analysis with NucleusJ plugin or so. https://www.biorxiv.org/content/10.1101/2021.11.26.470128v1

Small comments:

Line 85: „biosynthesis of cofactors“ – if cofactors are „non-protein chemical compound or metallic ion“, it can explain as biosynthesis. How metal ion can come from „biosynthesis“???

Line 247: „cell morphology“ – not cell morphology, but cell fate and epigenetic status.

Author Response

Point-to-point Response

Reviewer: 1

Comments and Suggestions for Authors

Thank you very much for response.

It is great that you plan to study cell-types specifci gene expression and histone modification. I am not sure methods like RNA-sequence in different tissues or single cells for further transcriptome analysis can help too much. The only way to clarify link between circadian clock and epigenetic is in situ by using, for example, antibody (https://www.thermofisher.com/antibody/product/H3K36me2-Antibody-Polyclonal/600-401-I87) and detailed chromatin analysis with NucleusJ plugin or so. https://www.biorxiv.org/content/10.1101/2021.11.26.470128v1

Response: Thank you so much for your helpful and kindly advice. It seems that NODeJ and NucleusJ are great plugins for detailed chromatin detection. Though it may be difficult for us to make Nuclear immunofluorescence images by H3K36me2-Antibody, we will try it as your suggestion from now. Thanks again.

Small comments:

Line 85: „biosynthesis of cofactors“ – if cofactors are „non-protein chemical compound or metallic ion“, it can explain as biosynthesis. How metal ion can come from „biosynthesis“???

Response: Sorry for making this confusing. “Metal ion” here means “Metal ionic compound”, such as Fe-coproporphyrin III. The whole pathway of the “biosynthesis of cofactors” can be found at https://www.genome.jp/pathway/ko01240.

Line 247: „cell morphology“ – not cell morphology, but cell fate and epigenetic status.

Response: Sorry for the inappropriate description, and we corrected it now following your comment. Thank you very much.

Round 3

Reviewer 1 Report

thank you very much!

You do not need to even use H3K36 antibody, even simple nuclei/chromatin anaylsis with these plugin can provide you a lot information for gene function and you will see very big difefrences in chromatin organization between cell types and cell posotion, what led to conclusion that total transcriptomic data is rather irrelevant.

I hope this will help you in future fir proper design of experiments.

Author Response

Reviewer: 1 (Round 3)

Comments and Suggestions for Authors

Thank you very much!

You do not need to even use H3K36 antibody, even simple nuclei/chromatin anaylsis with these plugin can provide you a lot information for gene function and you will see very big difefrences in chromatin organization between cell types and cell posotion, what led to conclusion that total transcriptomic data is rather irrelevant.

I hope this will help you in future fir proper design of experiments.

Response: Thank you very much for opening a new door for us. We appreciate all your help and comments.